

# DDoS attack detection in Edge-IIoT digital twin environment using deep learning approach

Feras Al-Obeidat[1], Adnan Amin[2], Ahmed Shuhaiber[1] and Inam ul Haq[2]

[1] College of Technological Innovation, Zayed University, Abu Dhabi, United Arab Emirates
[2] School of Computer Science and Information Technology, Institute of Management Sciences (IMSciences), Peshawar, Pakistan

## ABSTRACT

The industrial Internet of Things (IIoT) and digital twins are redefining how digital models and physical systems interact. IIoT connects physical intelligence, and digital twins virtually represent their physical counterparts. With the rapid growth of Edge-IIoT, it is crucial to create security and privacy regulations to prevent vulnerabilities and threats (*i.e.*, distributed denial of service (DDoS)). DDoS attacks use botnets to overload the target system with requests. In this study, we introduce a novel approach for detecting DDoS attacks in an Edge-IIoT digital twin-based generated dataset. The proposed approach is designed to retain already learned knowledge and easily adapt to new models in a continuous manner without retraining the deep learning model. The target dataset is publicly available and contains 157,600 samples. The proposed models M1, M2, and M3 obtained precision scores of 0.94, 0.93, and 0.93; recall scores of 0.91, 0.97, and 0.99; F1-scores of 0.93, 0.95, and 0.96; and accuracy scores of 0.93, 0.95, and 0.96, respectively. The results demonstrated that transferring previous model knowledge to the next model consistently outperformed baseline approaches.

## INTRODUCTION

In recent years, the Internet of Things (IoT) has advanced at a very fast rate due to its scalability, intelligence, and wide range of applications. IoT has taken a leading role in technology, particularly in comprehensive applications within different sectors and enterprises such as smart homes, smart cities, horticulture, transportation, healthcare, and even the armed forces (*Rai et al., 2023*). Equipment is connected to IoT networks, which communicate with and transfer data to consumers through the internet. In most IoT-based applications, there is very little or no interaction with human beings or physical objects; instead, tasks are performed automatically. Industrial Internet of Things (IIoT) is a subclass of IoT that is deployed in industrial settings to enhance productivity and drive efficiency in resource utilization. Generally, the technology sector has been primarily impacted by the IoT (*Ferrag et al., 2022*). The International Data Corporation estimated that by 2025, against the projected 8.1 billion people, there will be 41.6 billion internet-enabled devices that will generate 79.4 ZB of data (*Atlam & Wills, 2020*). Essentially, edge computing and IIoT can be effectively integrated into any industrial

Corresponding author
Adnan Amin,
adnan.amin@imsciences.edu.pk

application using digital twins—in other words, virtual models of physical assets, systems, or processes. The combination of digital twins with the IIoT offers substantial advantages to multiple sectors (*Kamath, Morgan & Ali, 2020*). Digital twins can be paired with edge computing to handle real-time data obtained from IIoT sensors and devices locally, decreasing latency for faster decision-making. This combination enables the optimization of industrial processes, predictive maintenance, and more efficient monitoring. The synergy resulting from the combination of digital twins and edge-IIoT creates a very potent framework for improving operational efficiency, reducing downtime, and facilitating seamless integration between different physical and digital systems operating in smart manufacturing environments (*Li et al., 2023*). On the other hand, with technological development, the number of attackers has risen. The unprecedented rise of cyber-attacks has significantly affected the economics of businesses that rely on computer networks (*Devan & Khare, 2020*). Various types of attacks have been used to cause network disruption. A denial-of-service (DoS) attack is the most common type of such attacks. Their frequency in the last 10 years has made them a potential threat to network stability, as they can disrupt numerous services (*Tao & Yu, 2013*). Distributed denial of service (DDoS) attacks reduce system performance and block legitimate access by launching significant attacks on the system simultaneously with a large network of infected computers (*Alomari et al., 2012*). Since its disclosure in September 2016 by the malware research group "Malware Must Die," the Mirai malware has been under scrutiny due to its role in malicious and devastating DDoS attacks. In the fourth quarter of 2016, the largest DDoS attack ever experienced occurred due to insecure IoT devices. The year 2016 is—and will continue to be—known as the year of Mirai. It took advantage of vulnerable IoT devices on October 21 to carry out the most powerful DDoS attack in history, reaching 1.2 terabits per second. DDoS attacks have been categorized into two types: volumetric attacks and application-layer attacks (*Adedeji, Abu-Mahfouz & Kurien, 2023*). To ensure the security of IoT/IIoT systems, it is essential to use datasets that accurately represent real-world IoT/IIoT applications. However, with the recent technology known as edge computing, as the number of edge devices increases, it becomes challenging to secure devices and networks against attacks. Edge IIoT is a contemporary cybersecurity dataset. A complex seven-layer testbed with more than 10 IoT devices, IIoT-based Modbus flows, and 14 protocol-related attacks were developed. This study describes the dataset and its properties (*Ferrag et al., 2022*). The goal of this study is to address the following research questions (RQs) through a systematic approach:

- **RQ1:** Can we develop an efficient technique for DDoS attack detection in Edge-IIoT devices within digital twin environments?
- **RQ2:** Can we construct a deep learning model that continuously learns and detects DDoS attacks in Edge-IIoT within digital twin environments?

The rest of this article is organized as follows: 'Literature review' presents the related studies followed by the proposed empirical evaluation setup in 'Empirical Evaluation Setup'. The results and discussion are presented in 'Results and Discussion', and the article is concluded in 'Conclusion'.

# LITERATURE REVIEW

This section starts with the definition of "DDoS attacks" and includes an in-depth review of previous studies on the subject.

## Denial of service as a cyber-attack

The DoS attack is a type of cyber-attack wherein a single host computer floods the server with several requests in order to make its services unavailable. In contrast, DDoS is a special type of DoS attack wherein a server is flooded with a huge volume of unwarranted requests that normally originate from a plethora of geographically dispersed devices known as botnets. The aim of this attack is to make services unavailable to valid traffic (*Khader & Eleyan, 2021*).

Four major ways through which DDoS attacks may take place can be distinguished: DDoS Internet Control Message Protocol (ICMP) attacks, DDoS UDP flood attacks, DDoS Transmission Control Protocol Synchronize (TCY-SYN) flood attacks, and DDoS HTTP attacks. In a DDoS ICMP attack, attackers overwhelm the system with ICMP echo requests, thereby crippling the services used by other applications on the victim's server. The attacker rapidly sends a continuous stream of echo requests to the victim to perform the attack. The DDoS UDP flood attack is a type of attack where the attacker floods the target system by sending a high volume of traffic consisting of UDP datagrams. Spoofed source IP addresses are used by this attack when launched from a single host. It can create a buffer overflow problem since the victim's server is overwhelmed with incoming datagrams. In this type of attack, the DDoS TCP SYN flood attack is one wherein fake SYN requests flood the victim's system using spoofed IP addresses. Since the IPs are fake, further responses to the victim's SYNchronize, SYNchronize-ACKnowledgement, and ACKnowledge (SYN/ACK) packets do not arrive, and hence the corresponding ports remain open unnecessarily. If many SYNs are generated, all the victim's ports become blocked, preventing real users from connecting. The DDoS HTTP attack sends a large number of fake HTTP requests to the target server from multiple random IP addresses. Since the IP addresses are spoofed, there is no initial setup for HTTP communication connection; it continuously engages the server in fake interactions for extended periods. As a result, it cannot accept connections from new, genuine users.

## Methods for detection of DDoS attacks in various domains

*Gurulakshmi & Nesarani (2018)* presented a model that effectively differentiates between normal and abnormal traffic flow in a network. They utilized support vector machines (SVM) and K-nearest neighbors (KNN) algorithms to accurately identify abnormal activity at an early stage. They utilized the XOIC tool to produce DDoS traffic, originating from multiple source addresses and targeting a single destination address. The packets were caught using Wireshark. Subsequently, they employed a packet analyzer to selectively sort the packets based on their respective protocols. Based on their experiment, the SVM algorithm achieved an accuracy of 95% while the KNN algorithm achieved an accuracy of 90% when applied to big feature datasets. For the dataset with less features, the SVM and KNN algorithms achieved accuracies of 97% and 98% respectively.

*Mohammed (2021)* used Decision Tree (DT), KNN, and naïve Bayes (NB) algorithms to classify benign network traffic from DDoS attacks. Nineteen different features were carefully selected from the CIC2019DDoS dataset. The experiment used several DDoS attack techniques including UDP, DNS, SYN, and NetBIOS. The results show that DT and KNN both achieved the best performance, at 100% and 98%, respectively. The result from the naïve Bayes algorithm was poor, with an accuracy rate of 29%. DT, KNN, and naïve Bayes accuracy rates were 100%, 96%, and 27%, respectively. Recall rates for DT, KNN, and NB were 100%, 97%, and 100%, respectively. The F1-scores for DT, KNN, and NB classifiers were 100%, 97%, and 42%, respectively.

*Ali et al. (2022)* proposed an Intrusion Detection System (IDS) that combines an improved genetic algorithm (GA) and backpropagation neural network (NN) with an autoencoder network model and an enhanced genetic algorithm (GA). This network is known as the IGA-BP network. With minimal processing complexity, the system obtained a detection rate of 98.98% and an accuracy of 99.29%. Using evolutionary sparse convolution networks and training patterns, the proposed IDS was able to tell the difference between normal and malicious IoT activities. They have used the dataset "CIC-DDoS2019." One of the constraints was the necessity to guarantee high reliability, rapid computation, and reduced complexity.

*Jiang et al. (2020)* introduced a PSO-XGBoost model to improve the parameters of XGBoost for the "NSL-KDD" dataset. The XGBoost model is applicable for addressing multi-classification problems. Particle swarm optimization (PSO) is effective in swiftly attaining a hypothesized optimal solution. The results show that their model has better mean average precision and macro values compared to other models like random forest (RF), Bagging, and AdaBoost. It had a 92% accuracy rate in detecting positive instances. All four models' performance curves for mean average precision are in line with one another; however, PSO-Xgboost stands out because it has the largest area under the curve, measuring 0.64.

*Ullah et al. (2024)* suggested an IDS with transformer-based transfer learning. Complex features and imbalanced data problems are challenging in network flows. They utilized the Synthetic Minority Oversampling Technique (SMOTE) technique to resolve the problem of imbalanced data in the "UNSW-NB15", "CIC-IDS2017", and "NSL-KDD" datasets. In this study, deep information from the balanced network flow is extracted using the convolutional neural network (CNN) model. Finally, based on those deep features, a hybrid convolutional neural networks and long short-term memory (CNN-LSTM) model is built to detect different attack types. Their proposed model achieved a high superiority of the baseline approaches with precision, recall, F1-score, and accuracy values of 99%, 100%, 99%, and 99.21%, respectively. Moreover, there is an experiment in explainable artificial intelligence (AI) to explain the devised methodology and also to explore the most dependable and effective qualities linked to certain assault types.

*Ikram et al. (2021)* in their research, used an ensemble of different deep neural network (DNN) models like MLP, BPN, and LSTM. The performance of the ensemble model was evaluated using two datasets: "UNSW-NB15" and "VIT SPARC20," both created on campus. The "VIT SPARC20" dataset contains all types of traffic: standard unencrypted

traffic, standard encrypted traffic, and both encrypted and unencrypted malicious traffic. The proposed models classify the encrypted normal and malicious traffic of "VIT SPARC20" without decrypting the contents. XGBoost boosts the performance by integrating the output from each deep learning model. The proposed model's maximum accuracy on the "UNSW NB" dataset is 99.5 percent, with precision 99.45 percent, recall 99.42 percent, and F1-score 99.5 percent. On "VIT SPARC20", it yielded an accuracy of 99.4% with 98% precision and 97% recall.

*Song et al. (2022)* presented the IDS using WOA-XGBoost on the "KDD CUP 99" dataset. The whale optimization algorithm is applied to obtain the best model parameter. First, the obtained network data was reduced in dimensionality by PCA. Then, it is fed to the WOA-XGBoost method to enhance the overall model accuracy of intrusion detection on data after training. They obtained 99% in accuracy, 99.5% sensitivity, and 95.4% in specificity.

*Alduailij et al. (2022)* suggested an efficient IDS with the XGBoost algorithm for feature selection and DNN to classify network intrusions. The various processes in the XGBoost-DNN model are normalization, feature selection, and classification. The softmax classifier was used in classifying network intrusions, while the Adam optimizer was used to improve learning rates during the training of the deep neural network. It was implemented in Python using TensorFlow on the NSL-KDD dataset. In order to validate the model, cross-validation was performed. The model attained 97% accuracy, precision, recall, and F1-score for a population size of 7,000.

*Elsayed et al. (2020)* proposed an approach to detect DDoS attacks in cloud computing with the aim of minimizing misclassification errors over the CICIDS 2017 and CICDDoS 2019 datasets. In this regard, two feature selection approaches, namely Mutual Information and Random Forest Feature Importance, were applied in order to retain the most relevant features. Further, the selected features feed the machine learning algorithms RF, GB, weighted voting ensemble, KNN, and LR for detecting DDoS attacks. Experimental results: RF, GB, WVE, and KNN achieve an accuracy of 99% using 19 characteristics. Moreover, misclassifications have also been analyzed in the article to enhance precision in the measurement, and performance of the RF classifier was outstanding.

*Dutta et al. (2020)* proposed "DDoSNet," a deep learning model for network attack detection. In their work, they used a deep learning approach along with a recurrent neural network and combined an autoencoder with a SoftMax regression model at the output layer that categorized network traffic into two classes: malicious and normal. The performance of their model was evaluated using the CICDDoS2019 dataset. Their model realized an accuracy as high as 99%. In normal and attack traffic, the accuracy was 100% and 99%, respectively. In both categories, the recall was 99%. For regular and attack traffic, the F1-score was 99%.

*Idrissi, Azizi & Moussaoui (2022)* have proposed the ensemble method for anomaly-based NIDS that incorporated the idea of layering generalization and deep learning algorithms. The research work, to achieve better efficiency, makes use of multiple feature engineering techniques with the help of dimensionality reduction. The performance is much improved through incorporation with a meta-classifier along with

**Table 1 Summary of various intrusion detection approaches.**

| Ref. | Technique used | Dataset | Outcomes |
|---|---|---|---|
| *Tao & Yu (2013)* | CNN, pruning, quantization clustering | MQTTIoT- IDS2020 | Achieved accuracy 97.74% after pruning, quantization and clustering. |
| *Ullah et al. (2024)* | DNN and Pruning | KDDCPU 99 | Obtained accuracy 93.71% after applying DNN and Pruning. |
| *Sharmila & Nagapadma (2023)* | Autoencoder, QAE-unit8 | RTIoT 23 | Received accuracy 96.35% for Autoencoder and QAE-unit8. |
| *Maulana Ibrahimy, Dewanta & Erza Aminanto (2022)* | Random forest, Decision tree, KNN | InSDN | Based on the feature correlation and elimination achieved accuracy 99%. |
| *Farahani (2020)* | Decision tree | KDD Cup, 99NSL and CIC IDS'17 | Achieved various accuracies *i.e.*, 85.19% to 98.42%. |

DNN and LSTM networks. In this experiment, they used the "IoT-23", "LITNET-2020", and "NetML-2020" datasets. Utilizing the "NetML-2020" dataset, they reached a detection accuracy of 100%; by the "LITNET" dataset, they achieved an accuracy of 100%; using the "IoT-23", they reached 99.7% for the IoT-23 dataset. They attained an F1-score of as high as 98%, recall of 95%, and precision of 100%. Precision, recall, and F1-score were all equal to 100% for the "LITNET" dataset. Precision was 99%, recall was 99.9%, and F1-score was 100% on the "NetML-2020" dataset. There are two major shortcomings even though their model achieved the maximum accuracy up to 100%. The elapsed time of the techniques was not discussed. Then, stacking was used as the only ensemble technique. As such, the performance of the model is to be evaluated by testing and experimenting with various ensemble techniques. The following Table 1 compares various intrusion detection approaches (see *Tao & Yu, 2013*; *Ullah et al., 2024*; *Sharmila & Nagapadma, 2023*; *Maulana Ibrahimy, Dewanta & Erza Aminanto, 2022*; *Farahani, 2020*).

To the best of our knowledge, there is a lack of strategy-based continuous deep learning studies concerning intrusion detection in Edge-IIoT in a digital twin environment. This motivated us to review this domain with more attention and shed light on these approaches. In particular, the absence of a unified framework that can perpetually learn and unlearn develops unresilient and inflexible models that cannot dynamically learn and forget. Such a nonexistence of disjoint approaches to handle cases in which a model needs to evolve from knowledge relevant yet outdated or even sensitive has confined their applicability in digital twin environments. Thus, we are proposing the controlled knowledge distillation framework that would bridge the gap and be able to support continual as well as deep learning.

## EMPIRICAL EVALUATION SETUP

This section presents a general framework of the setup for the evaluation and the proposed approach's effectiveness towards DDoS attacks based on the usage of continuous deep learning. The following subsections provide a detail of the dataset, pre-processing steps before the experiment, the proposed framework, model architectures, and the design of the problem being addressed.

**Table 2 Description of the samples and their counts against the class labels.**

| Labels | Samples |
| --- | --- |
| Normal | 24,101 |
| DDoS UDP | 14,498 |
| DDoS ICMP | 14,090 |
| Ransomware | 10,925 |
| DDoS HTTP | 10,561 |
| SQL injection | 10,311 |
| Uploading | 10,269 |
| DDoS TCP | 10,247 |
| Backdoor | 10,195 |
| Vulnerability scanner | 10,076 |
| Port scanning | 10,071 |
| XSS | 10,052 |
| Password | 9,989 |
| MITM | 1,214 |
| Fingerprinting | 1,001 |

## Edge-IIoT digital twin testbed framework

While datasets utilized by cybersecurity researchers are often proprietary or open-source and not field-specific, because the number of IIoT datasets available is too few, researchers can still acquire such attributes from the dependencies of IIoT applications on existing ones. Our study considers a realistic testbed (*Ferrag et al., 2022*) that mirrors an actual IIoT environment. This digital-twin environment enables real-time network traffic monitoring by continuously synchronizing with edge devices. Various realistic intrusions were simulated to collect data sets comprising legitimate and malicious traffic. The seven-layer testbed includes cloud computing, network functions virtualization (NFV), blockchain, fog, SDN, edge, and IIoT perception layers. For DDoS detection, we leverage key digital twin characteristics, such as state synchronization, real-time anomaly detection, and automated threat response mechanisms. These characteristics allow our approach to identify abnormal traffic patterns indicative of DDoS attacks while continuously adapting to emerging threats. As we are only concerned with DDoS attack detections, we have focused on those records that relate to DDoS attacks (*e.g.*, DDoS UDP, ICMP, ransomware, DDoS HTTP, SQL injection). The subject data sets can be obtained from the source of the publicly available data set (*Ferrag et al., 2022*). Table 2 reflects the description and count of the samples of the various class labels.

## Data wrangling

The dataset consists of 157,000 structured network traffic samples, each represented by multiple numerical and categorical features describing packet-level and flow-level behavior. Before training the models, several preprocessing steps were carried out to ensure the data was clean, consistent, and properly formatted for use with deep learning models. The original dataset contained 63 features, representing diverse aspects of network

traffic such as packet-level metadata, protocol flags, content fields, and flow identifiers. The descriptive statistics of the numerical and categorical variables in the original dataset are presented in Tables A1 and A2, respectively.

To enhance the model's learning efficiency and eliminate irrelevant or noisy features, we applied domain-informed feature selection (*Ferrag et al., 2022*). We dropped unnecessary features prior to the model training process based on the following considerations: (i) high cardinality features such as frame.time, ip.src_host, ip.dst_host, arp.src.proto_ipv4, and arp.dst.proto_ipv4 were removed due to their high uniqueness across samples. These fields typically act as identifiers or session-specific values with minimal generalization capacity and high potential to introduce noise or overfitting; (ii) Unstructured or Raw Content: fields like http.file_data, tcp.payload, and mqtt.msg contain raw data (*e.g.*, payloads or decoded message content) not easily parsable into meaningful numerical features without deep packet inspection or NLP-based preprocessing; (iii) Low Variability or Redundancy: certain protocol-related fields (*e.g.*, http.request.method, http.request.version, dns.qry.type, and icmp.unused) showed either no variability (zero standard deviation) or negligible contribution to classification in exploratory analysis; (iv) Port and Payload Metadata: features such as tcp.srcport, tcp.dstport, and udp.port often vary randomly in benign traffic and are frequently reused in malicious flows, making them inconsistent discriminators of attack behavior in IIoT environments. Similarly, features like tcp.options, dns.retransmit_request_in, and mqtt.protoname were found to be either redundant or sparsely populated.

The complete list of dropped features is as follows: {"frame.time, ip.src_host, ip.dst_host, arp.src.proto_ipv4, arp.dst.proto_ipv4, http.file_data, http.request.full_uri, icmp.transmit_timestamp, http.request.uri.query, tcp.options, tcp.payload, tcp.srcport, tcp.dstport, udp.port, mqtt.msg, icmp.unused, http.tls_port, dns.qry.type, dns.retransmit_request_in, mqtt.msg_decoded_as, mbtcp.trans_id, mbtcp.unit_id, http.request.method, http.referer, http.request.version, dns.qry.name.len, mqtt.conack.flags, mqtt.protoname, mqtt.topic"}.

After feature elimination, the list of remaining features are the following such as {"arp.opcode, arp.hw.size, icmp.checksum, icmp.seq_le, http.content_length, http.response, tcp.ack, tcp.ack_raw, tcp.checksum, tcp.connection.fin, tcp.connection.rst, tcp.connection.syn, tcp.connection.synack, tcp.flags, tcp.flags.ack, tcp.len, tcp.seq, udp.stream, udp.time_delta, dns.qry.name, dns.qry.qu, dns.retransmission, dns.retransmit_request, mqtt.conflag.cleansess, mqtt.conflags, mqtt.hdrflags, mqtt.len, mqtt.msgtype, mqtt.proto_len, mqtt.topic_len, mqtt.ver, mbtcp.len"}.

Due to the presence of extreme values and inconsistencies in the scale of certain features, we employed the normalization process through RobustScaler, which scales the data using the median and interquartile range (IQR), making it more resilient to outliers than standard z-score normalization. This technique has been effectively used in cybersecurity datasets where outlier resistance is crucial for stable model training (*Aggarwal & Yu, 2015*). Further, we analyzed that there are a total of 5,404 samples that are redundant in the target dataset out of 157,800 samples. All duplicated records were eliminated and not considered for further experiments.

**Table 3 Labels encoding for binary class labels.**

| Encoded labels | Description samples | Count |
|---|---|---|
| 0 | Normal | 24,101 |
| 1 | Attacks | 128,095 |

Although the dataset consists of tabular data, we aimed to use a 1D CNN to capture local feature patterns. To enable this, the feature matrix was reshaped from a 2D array (*i.e.*, samples × features) to a 3D array of shape (samples, features, 1). This transformation prepares the input for Conv1D layers, which expect temporal-like or sequential input structures. Previous studies have shown that when features are organized in a consistent order, Conv1D models can effectively extract inter-feature dependencies in intrusion detection tasks (*Ullah et al., 2024*; *Dutta et al., 2020*). The ability of CNNs to capture feature relationships through convolutional layers makes them well-suited for learning attack patterns, even when individual samples do not represent sequential time-series data.

Finally, we obtained a total of 152,196 distinct samples where 24,101 samples belonged to the normal class label and 128,095 samples were a combination of the remaining class labels. Further, we followed the procedure from the study (*Amin et al., 2024*) and transformed the multi-class problem into a binary classification problem. Additionally, the label encoding method was applied to the binary class data and the class samples are represented as given in Table 3.

## Data balancing

To address the class imbalance problem in the target dataset, we applied the SMOTE algorithm. After applying SMOTE, the sample distribution of all classes was balanced where minority class samples were oversampled to equal the majority class samples. Table 4 reflects the distribution of the class samples.

## Problem structure

In this section, the problem structure is described to understand the underlying problem before presenting the proposed approach to detect DDoS attacks in digital twin environments in such a way that prior knowledge of the detection system can be incorporated with knowledge learned from the new model on new unseen data to enhance overall detection capability in an efficient manner. As stated previously, this study describes the proposed continuous learning approach for the detection of DDoS attacks using a deep neural network in continuous fashion.

The purpose of this study is to develop a framework that extends the knowledge of a baseline model without training on previous data points and by incorporating new learnings from a new model to predict a class label (0 = Normal or 1 = Attacks) y = 0, 1 for each set of input attributes $A \in \{a_1, a_2, a_3, \ldots, a_n\}$, at any stage of the model process. An example is a pair $(X, y)$; for instance, X is a set of attributes that have values for activities or operations performed in the testbed in digital twin environments, and each activity contains 61 attributes, excluding the class label.

**Table 4  Number of samples before and after SMOTE.**

| Encoded labels | Target labels | Sample size before oversampling | Sample size after oversampling | difference |
|---|---|---|---|---|
| 0 | Normal | 24,101 | 128,095 | 103,994 |
| 1 | Attacks | 128,095 | 128,095 | 0 |

## Construction of continuous deep learning approach

The CNN is used as baseline classifier for the proposed study. In CNNs, the hidden layers perform convolutions, which are mathematical operations defined by Eq. (1).

$$s(t) = (f * g)(t) = \int f(\tau)g(t-\tau)\,d\tau. \tag{1}$$

At a specified time t, the function $g(t-\tau)$ gives more importance to local observations than distant ones. The weighted average of the input $f(\tau)$ is calculated. Equation (2) gives the discrete version of the equation above.

$$s(t) = (f * g)(t) = \sum_{\tau} f(\tau)g(t-\tau). \tag{2}$$

By summing the products of the functions $f(\tau)$ and $g(t-\tau)$, Eq. (2) can determine the output value at discrete time t for each possible value of $\tau$.

CNNs are generally composed of one or more of the following layers: (i) convolutional layers—this layer forms the hierarchical core of the network. At this point, it extracts several features from the input data. It processes the input data using an N × N filter. As it moves along with the input data, the filter calculates the dot product between itself and the relevant parts, considering its size (N × N); (ii) max-pooling layers—CNN models have pooling layers in addition to convolutional layers. The pooling layer compresses the output of a convolutional layer to reduce feature dimensions that are convolved over and consequently reduce computation. This is achieved through processing individual feature maps and removing connections between layers. The methodologies differ based on varying methods. The intermediate layer connects the convolutional layer to the fully connected (FC) layer. The pooling layer is positioned strategically after the convolutional layer. The main purpose of the pooling layer aligns with our objective of continuous learning; and (iii) fully connected layers—this level takes the input obtained from the previous layer in one dimension and passes it to the fully connected layer. The flattened vector is then passed through fully connected layers for mathematical operations. Classification takes place at this stage. Generally, two interconnected fully connected layers give better results than one connected layer. Thus, two connected layers with interconnection work well. This also reduces human supervision across the entire process of CNN layers.

The development and training of a series of deep learning models involving Base Model, Model B, and Model C are to be conducted based on the completion of preprocessing. The explicit division of data into separate sections for training, validation, and testing is an important choice that has significant impacts on the reliability and results of the constructed models in machine learning (ML) and data-driven research. A highly strict

**Table 5 Distribution of the data for models (base, B, and C).**

| Encoded labels | Model | Sample size |
| --- | --- | --- |
| 0 | Base Model (M1) | 32,023 |
| 1 | Base Model (M1) | 32,023 |
| 0 | Model B (M2) | 35,866 |
| 1 | Model B (M2) | 35,866 |
| 0 | Model C (M3) | 40,989 |
| 1 | Model C (M3) | 40,989 |
| 0 | Test set | 19,213 |
| 1 | Test set | 19,213 |

data partitioning method is used in this study to enable the building and validation of three unique proposed models, namely Base Model, Model B, and Model C.

The concept of data distribution according to proportional representation is the foundation upon which all data partitioning techniques are based. The available dataset is divided into four strictly segregated subgroups, each performing a different role, to ensure fair provisioning of data for purposes such as training, validation, and testing. The proportions selected for this allocation are as follows: (i) the base model contains 25% of all data and is therefore the largest contributor to the dataset. This component of the data is the principal dataset that will be used to train and validate the proposed baseline model, "Base Model". Due to its large size, the model receives enough data to understand some correlations and patterns that are developing; (ii) Model B contains 28% of the overall data. This split aids Model B in extracting details from the dataset since it allows iterative improvement of the model; and (iii) Model C is given a different subset, which constitutes 32% of the data, similar to Model B. The subset makes it easier to train and test Model C on new tasks so that Model C can learn the characteristics of a particular section of the data it is processing; finally, (iv) the remaining 15% of the data is kept as a test set. This testing set is crucial, as it serves as the neutral standard based on which the capabilities of each of the three models can be compared. A fair and unbiased judgment of the performance of different models is highly needed. One way to ensure this is to use the same testing set for all models. Table 5 represents the prepared dataset according to the data distribution against the target class labels.

## Configuration of the base model

Initially, we trained the base model (M1) on the specified portion of data with the following CNN model configuration, as given in Table 6. The total number of parameters in the model was recorded as approximately 36,225, where trainable and non-trainable parameters were 36,097 and 128, respectively.

In the next section, we discuss all the results observed from a series of experiments.

## RESULTS AND DISCUSSION

This section presents a comprehensive discussion on the obtained results from various experiments.

**Table 6 Base model (M1) configuration.**

| Layer (type) | Output shape | Param # |
| --- | --- | --- |
| Conv1D | (None, 33, 128) | 512 |
| MaxPooling1D | (None, 11, 128) | 0 |
| Dropout | (None, 11, 128) | 0 |
| Conv1D | (None, 11, 64) | 24,640 |
| Batch normalization | (None, 11, 64) | 256 |
| Max pooling1d | (None, 5, 64) | 0 |
| Dropout | (None, 5, 64) | 0 |
| Flatten | (None, 320) | 0 |
| Dense | (None, 32) | 10,272 |
| Dropout | (None, 32) | 0 |
| Flatten | (None, 32) | 0 |
| Dense | (None, 16) | 228 |
| Dense | (None, 1) | 17 |

## Experiment # 1: Base Model (M1)

In the first experiment, the CNN architecture with activation function "ReLU", epochs set to 25, learning rate fine-tuned and set to 0.0004, and Adam used as optimizer in the training of the M1. The following layers were used in training this M1: (i) convolutional layers for feature extraction; (ii) max-pooling layers to reduce feature set dimensionality; and (iii) dropout layers as preventers of overfitting. Figure 1 illustrates the M1 validation accuracy and loss values in Figs. 1A and 1B, respectively.

Initially, the M1 had no previous model knowledge. Therefore, the base model was initially trained on a specific portion of data. Then the base model was evaluated on the validation and test data as shown in Table 7. M1 obtained over 0.93 accuracy on the validation data as shown in Fig. 1A. Similarly, we illustrated the validation loss score of M1 parallel to the accuracy in Fig. 1B. However, we validated the model performance on the test set, and the M1 obtained precision, recall, F1-score, and accuracy scores of 0.94, 0.91, 0.93, and 0.93, respectively, with model M1.

## Experiment # 2: Model B (M2)

In the second experiment, we initially provided the knowledge obtained from model M1 to M2 (knowledge transfer), and we froze the transferred layers in model M2 because we fully utilized the model M1 layers. In Model B, after freezing the layers inherited from the base model, a new dense layer with 128 units and a rectified linear unit (ReLU) activation function was added. This layer ensures that the M2 can achieve task-specific learning on new incoming data without having to retrain the base model.

On the other hand, we updated the weights of the neurons with previous weights. The activation function "ReLU" and flatten layers are applied to M2. The flatten layer converts the multi-dimensional output of the previous layer into a one-dimensional array. The M2 model is designed like M1 with activation function "ReLU", epochs set to 25, learning rate 0.0004, and Adam used as an optimizer in the training of model M2. Furthermore,

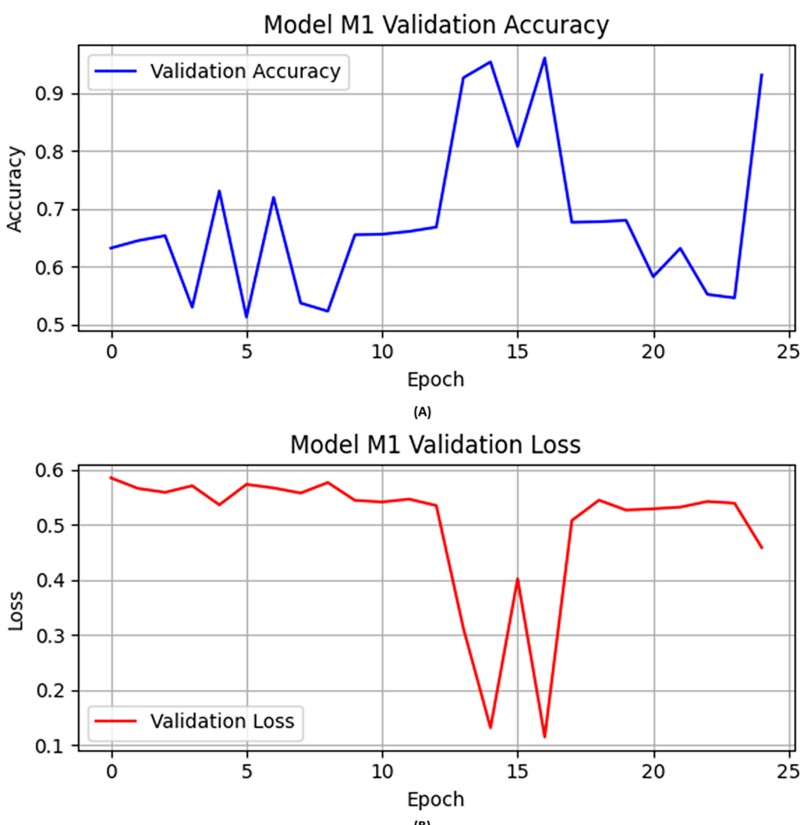

**Figure 1** The base model (A) validation accuracy, and (B) loss function results.

**Table 7 Performance summary of the M1.**

| Precision | Recall | F1-score | Accuracy |
|---|---|---|---|
| 0.94 | 0.91 | 0.93 | 0.93 |

dropout layers are also used after the flatten layer and dense layer. A dropout layer is integrated following the new dense layer to mitigate overfitting, with a dropout rate of 0.3. Figure 2 depicts the visual representation of the M2 validation accuracy and validation loss.

The CNN deep layers have their optimal weights for each neuron calculated. This results in the M2 model learning from the previous model M1 and then being trained further on a separate set of data meant for M2. This shows that the model has learned new samples without requiring full retraining on all accumulated data so far. The base model was then evaluated on the validation and test data. The performance of M2 is shown in Table 8 in terms of precision, recall, F1-score, and accuracy scores of 0.93, 0.97, 0.95, and 0.95, respectively, while the accuracy obtained on the validation data is shown in Fig. 2A. Similarly, we illustrated the validation loss score of M2 parallel to the accuracy in Fig. 2B. However, we validated the model performance on the test set, and the model obtained 0.95 accuracy.

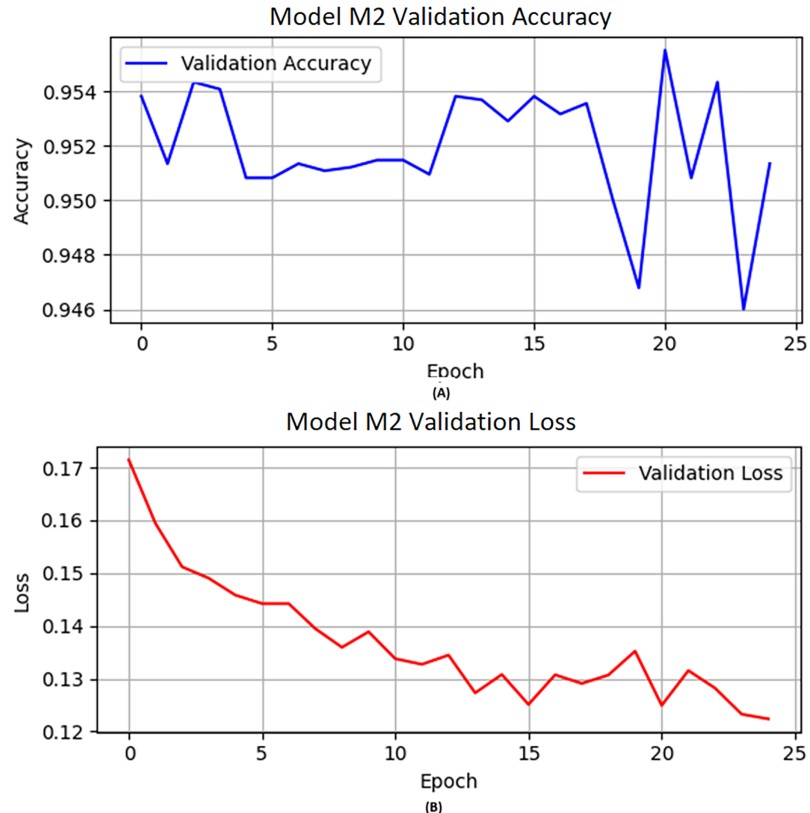

**Figure 2** The M2 model (A) validation accuracy and (B) loss function results.

**Table 8 Performance summary of the M2.**

| Precision | Recall | F1-score | Accuracy |
|---|---|---|---|
| 0.93 | 0.97 | 0.95 | 0.95 |

### Experiment # 3: Model C (M3)

M3 obtained knowledge from the previously trained model M2 and was further trained on a specified number of new samples. Furthermore, combined with optimization technique "Adam", learning rate 0.0004, epochs set to 50, and activation function "ReLU" used for M3. A similar approach was used as we applied previously for M2. Figure 3 describes the visual representation of the M3 validation accuracy and validation loss.

Table 9 reflects the performance of M3 in terms of precision, recall, F1-score, and accuracy scores of 0.93, 0.99, 0.96, and 0.96, respectively, while accuracy on the validation data is shown in Fig. 3A. Similarly, we illustrated the validation loss score of M3 parallel to the accuracy in Fig. 3B. However, we validated the model performance on the test set, and the model obtained 0.96 accuracy with model M3.

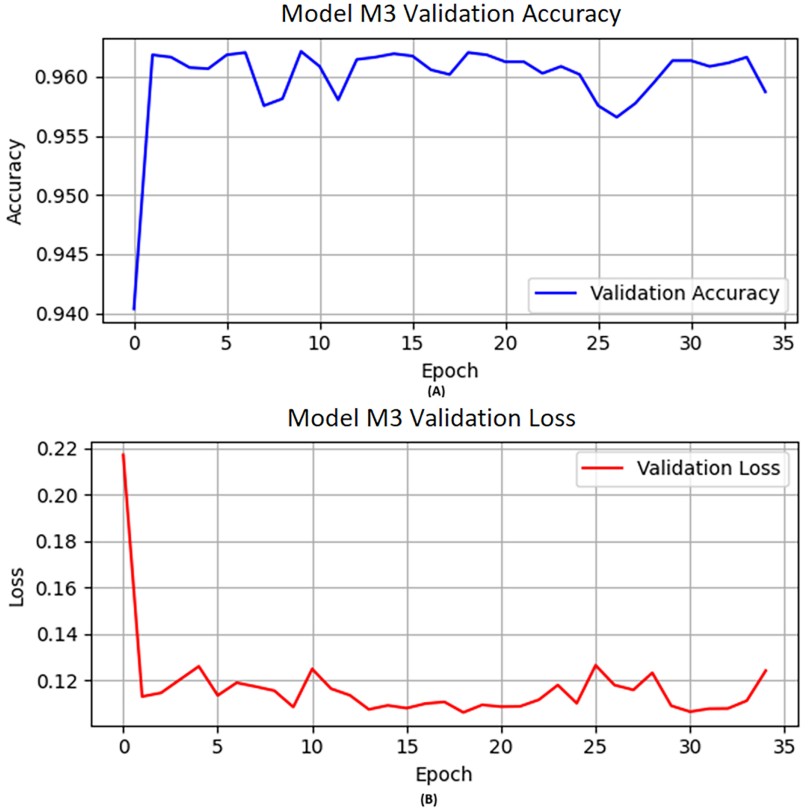

**Figure 3** The M3 model (A) validation accuracy and (B) loss function results.

**Table 9 Performance summary of the M3.**

| Precision | Recall | F1-score | Accuracy |
| --- | --- | --- | --- |
| 0.93 | 0.99 | 0.96 | 0.96 |

**Table 10 Performance summary of all models.**

| Models | Precision | Recall | F1-score | Accuracy |
| --- | --- | --- | --- | --- |
| M1 | 0.94 | 0.91 | 0.93 | 0.93 |
| M2 | 0.93 | 0.97 | 0.95 | 0.95 |
| M3 | 0.93 | 0.99 | 0.96 | 0.96 |

Furthermore, we compared the obtained results from all three experiments in which each model's knowledge is transferred to the next model along with its own knowledge obtained from additional new data, and the model learning process is iterative and continuous in nature. Table 10 describes the overall summary of the obtained performance by all three models. Figure 4 is a visual demonstration of all the models' performance.

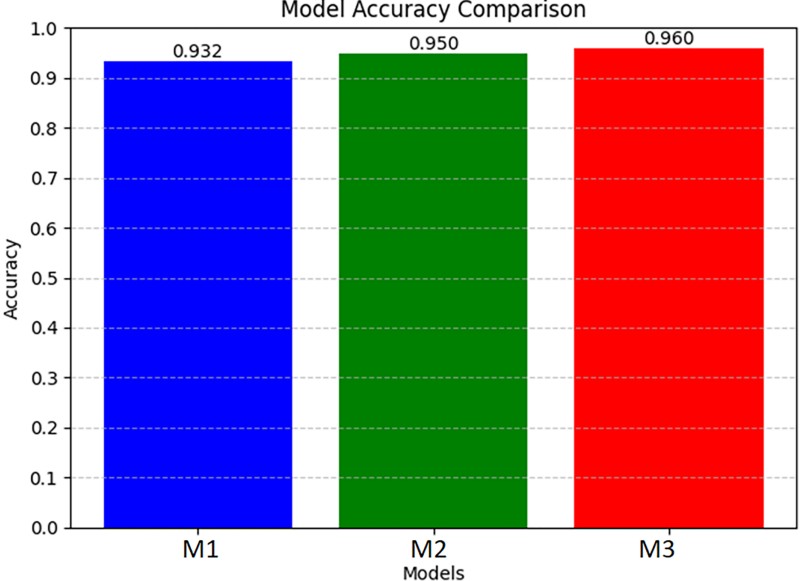

**Figure 4 Illustrates the performance of the M1, M2, and M3.**

## THREATS TO VALIDITY

While this study demonstrates promising results in detecting DDoS attacks in an Edge-IIoT environment using continual deep learning, several factors may influence its validity:

- Generalizability of results: This study's dataset is based on a simulated digital twin testbed. Although controlled and reproducible experiments are ensured, deployment in the real world could bring unintended network variations.
- Computational overhead: Ongoing learning avoids a complete retraining of the model, reducing computational expense substantially. Even so, there is still the potential for latency issues with processing intricate deep learning models at the edge. Model efficiency optimization *via* lean architecture and hardware acceleration is an area ripe for future research.

## CONCLUSION

In this study, we presented a deep learning-based continual learning framework using CNN models (M1, M2, and M3) for detecting DDoS attacks in an Edge-IIoT digital twin environment. Each model incrementally incorporated knowledge from the preceding one, demonstrating progressive improvement in performance. Our conclusions are grounded in a comprehensive understanding of the dataset's statistical properties. Prior to training, we conducted detailed data preprocessing—including removal of low-variance and redundant features and normalization of feature values—to ensure model robustness and eliminate noise. These steps were crucial in addressing the challenges posed by uneven feature scales and irrelevant attributes that could otherwise impair learning. Although

CNNs are traditionally associated with image or spatial data, we leveraged their capacity to capture local dependencies within structured feature sets by reshaping the input into a fixed-size representation. This approach allowed the convolutional layers to extract meaningful patterns from feature correlations, enhancing the detection accuracy even in tabular network traffic data. Our empirical results support this choice, with the final model (M3) achieving a precision of 0.93, recall of 0.99, F1-score of 0.96, and accuracy of 0.96 on the test set.

This work highlights the viability of CNN-based architectures for structured cybersecurity data and introduces a scalable methodology for continuous adaptation in real-time IIoT environments. Future work will focus on validating the model across more diverse, real-world datasets to further assess generalizability and performance under varying network conditions.

## APPENDIX

### Description of original dataset (numerical features only)

#### Table A1

| Name | Count | Mean | Std | Min | 25% | 50% | 75% | Max |
| --- | --- | --- | --- | --- | --- | --- | --- | --- |
| arp.opcode | 157,800.0 | 1.419518e−02 | 1.497828e−01 | 0.0 | 0.0 | 0.000000e+00 | 0.000000e+00 | 2.000000e+00 |
| arp.hw.size | 157,800.0 | 5.984791e−02 | 5.962449e−01 | 0.0 | 0.0 | 0.000000e+00 | 0.000000e+00 | 6.000000e+00 |
| icmp.checksum | 157,800.0 | 3.047292e+03 | 1.114433e+04 | 0.0 | 0.0 | 0.000000e+00 | 0.000000e+00 | 6.553200e+04 |
| icmp.seq_le | 157,800.0 | 3.239980e+03 | 1.140607e+04 | 0.0 | 0.0 | 0.000000e+00 | 0.000000e+00 | 6.552400e+04 |
| icmp.transmit_timestamp | 157,800.0 | 4.046816e+04 | 1.764075e+06 | 0.0 | 0.0 | 0.000000e+00 | 0.000000e+00 | 7.728902e+07 |
| icmp.unused | 157,800.0 | 0.000000e+00 | 0.000000e+00 | 0.0 | 0.0 | 0.000000e+00 | 0.000000e+00 | 0.000000e+00 |
| http.content_length | 157,800.0 | 1.471552e+01 | 2.296597e+02 | 0.0 | 0.0 | 0.000000e+00 | 0.000000e+00 | 8.365500e+04 |
| http.response | 157,800.0 | 4.574778e−02 | 2.089383e−01 | 0.0 | 0.0 | 0.000000e+00 | 0.000000e+00 | 1.000000e+00 |
| http.tls_port | 157,800.0 | 0.000000e+00 | 0.000000e+00 | 0.0 | 0.0 | 0.000000e+00 | 0.000000e+00 | 0.000000e+00 |
| tcp.ack | 157,800.0 | 7.160039e+07 | 3.101231e+08 | 0.0 | 0.0 | 1.000000e+00 | 4.790000e+02 | 2.147333e+09 |
| tcp.ack_raw | 157,800.0 | 1.358347e+09 | 1.295523e+09 | 0.0 | 0.0 | 1.160051e+09 | 2.372228e+09 | 4.294947e+09 |
| tcp.checksum | 157,800.0 | 2.579660e+04 | 2.151303e+04 | 0.0 | 2,982.0 | 2.390600e+04 | 4.473300e+04 | 6.553500e+04 |
| tcp.connection.fin | 157,800.0 | 5.814322e−02 | 2.340148e−01 | 0.0 | 0.0 | 0.000000e+00 | 0.000000e+00 | 1.000000e+00 |
| tcp.connection.rst | 157,800.0 | 9.411914e−02 | 2.919953e−01 | 0.0 | 0.0 | 0.000000e+00 | 0.000000e+00 | 1.000000e+00 |
| tcp.connection.syn | 157,800.0 | 1.278517e−01 | 3.339257e−01 | 0.0 | 0.0 | 0.000000e+00 | 0.000000e+00 | 1.000000e+00 |
| tcp.connection.synack | 157,800.0 | 2.994930e−02 | 1.704480e−01 | 0.0 | 0.0 | 0.000000e+00 | 0.000000e+00 | 1.000000e+00 |
| tcp.dstport | 157,800.0 | 1.796465e+04 | 2.415422e+04 | 0.0 | 80.0 | 1.883000e+03 | 4.549400e+04 | 6.553500e+04 |
| tcp.flags | 157,800.0 | 1.261400e+01 | 9.319136e+00 | 0.0 | 2.0 | 1.600000e+01 | 2.000000e+01 | 2.500000e+01 |
| tcp.flags.ack | 157,800.0 | 6.352471e−01 | 4.813623e−01 | 0.0 | 0.0 | 1.000000e+00 | 1.000000e+00 | 1.000000e+00 |
| tcp.len | 157,800.0 | 1.297793e+02 | 1.307038e+03 | 0.0 | 0.0 | 0.000000e+00 | 1.400000e+01 | 6.522800e+04 |
| tcp.seq | 157,800.0 | 1.875111e+06 | 1.579707e+07 | 0.0 | 0.0 | 1.000000e+00 | 1.190000e+02 | 2.079647e+08 |
| udp.port | 157,800.0 | 7.748479e+00 | 6.134448e+02 | 0.0 | 0.0 | 0.000000e+00 | 0.000000e+00 | 6.031000e+04 |
| udp.stream | 157,800.0 | 1.211405e+05 | 4.687607e+05 | 0.0 | 0.0 | 0.000000e+00 | 0.000000e+00 | 2.898725e+06 |
| udp.time_delta | 157,800.0 | 3.414068e−01 | 9.686192e+00 | 0.0 | 0.0 | 0.000000e+00 | 0.000000e+00 | 5.070000e+02 |

(Continued)

| Table A1 (continued) | | | | | | | | |
| --- | --- | --- | --- | --- | --- | --- | --- | --- |
| Name | Count | Mean | Std | Min | 25% | 50% | 75% | Max |
| dns.qry.name | 157,800.0 | 1.270061e+04 | 1.568478e+05 | 0.0 | 0.0 | 0.000000e+00 | 0.000000e+00 | 2.896968e+06 |
| dns.qry.qu | 157,800.0 | 7.786692e−01 | 2.306341e+01 | 0.0 | 0.0 | 0.000000e+00 | 0.000000e+00 | 1.028000e+03 |
| dns.qry.type | 157,800.0 | 0.000000e+00 | 0.000000e+00 | 0.0 | 0.0 | 0.000000e+00 | 0.000000e+00 | 0.000000e+00 |

## Description of the original dataset (Categorical and Objects datatype only)

| Table A2 Summary statistics. | | | | |
| --- | --- | --- | --- | --- |
| Field | Count | Unique | Freq | Top |
| frame.time | 157,800 | 155,186 | 1,402 | 192.168.0.128 |
| ip.src_host | 157,800 | 19,090 | 72,546 | 192.168.0.128 |
| ip.dst_host | 157,800 | 8,084 | 75,373 | 192.168.0.128 |
| arp.dst.proto_ipv4 | 157,800 | 8 | 153,610 | 0 |
| arp.src.proto_ipv4 | 157,800 | 8 | 140,514 | 0 |
| http.file_data | 157,800 | 496 | 117,122 | 0.0 |
| http.request.uri.query | 157,800 | 1,665 | 137,413 | 0.0 |
| http.request.method | 157,800 | 6 | 96,542 | 0.0 |
| http.referer | 157,800 | 4 | 127,111 | 0.0 |
| http.request.full_uri | 157,800 | 4,073 | 96,542 | 0.0 |
| http.request.version | 157,800 | 8 | 95,328 | 0.0 |
| tcp.options | 157,800 | 73,139 | 50,728 | 0.0 |
| tcp.payload | 157,800 | 27,369 | 75,013 | 0 |
| tcp.srcport | 157,800 | 32,186 | 34,437 | 80.0 |
| dns.qry.name.len | 157,800 | 8 | 133,272 | 0.0 |
| mqtt.conack.flags | 157,800 | 3 | 133,499 | 0.0 |
| mqtt.msg | 157,800 | 117 | 133,499 | 0.0 |
| mqtt.protoname | 157,800 | 3 | 133,499 | 0.0 |
| mqtt.topic | 157,800 | 3 | 133,499 | 0.0 |
| Attack_type | 157,800 | 15 | 24,301 | Normal |

### Funding

This work was funded by a grant from Zayed University (Grant No. R22049). The funders had no role in study design, data collection and analysis, decision to publish, or preparation of the manuscript.

### Grant Disclosures

The following grant information was disclosed by the authors:
Zayed University: R22049.

## Competing Interests

The authors declare that they have no competing interests.

## Author Contributions

- Feras Al-Obeidat conceived and designed the experiments, performed the experiments, analyzed the data, authored or reviewed drafts of the article, and approved the final draft.
- Adnan Amin conceived and designed the experiments, performed the experiments, analyzed the data, performed the computation work, prepared figures and/or tables, latex code, and approved the final draft.
- Ahmed Shuhaiber analyzed the data, prepared figures and/or tables, and approved the final draft.
- Inam ul Haq conceived and designed the experiments, performed the experiments, performed the computation work, authored or reviewed drafts of the article, and approved the final draft.

## Data Availability

The Edge-IIoTset Cyber Security Dataset of IoT & IIoT is available at Kaggle: https://www.kaggle.com/datasets/mohamedamineferrag/edgeiiotset-cyber-security-dataset-of-iot-iiot.

The code is available in the Supplemental File.

## Supplemental Information

Supplemental information for this article can be found online at http://dx.doi.org/10.7717/peerj-cs.3052#supplemental-information.

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
