# Peer review of "DDoS attack detection in Edge-IIoT digital twin environment using deep learning approach"

_PeerJ Computer Science, doi:10.7717/peerj-cs.3052_

## Round 0.1 · original submission · Major Revisions

I have received reviews of your manuscript from scholars who are experts on the cited topic. They find the topic interesting; however, several concerns regarding dataset description (features), experimental setup, and comparison with current approaches must be addressed. These issues require a major revision. Please refer to the reviewers’ comments at the end of this letter; you will see that they advise you to revise your manuscript. If you are prepared to undertake the work required, I would be pleased to reconsider my decision. Please submit a list of changes or a rebuttal against each concern when you submit your revised manuscript.

Thank you for considering PeerJ Computer Science for the publication of your research.

With kind regards,

Reviewer 1 ·

Basic reporting

No comment

Experimental design

No Comment

Validity of the findings

No

Additional comments

The title of the article is "DDoS attack detection in Edge-IIoT digital twin environment using deep learning
approach".
crucial to create security and privacy regulations to prevent vulnerabilities and threats
(i.e., DDoS). DDoS attacks use botnets to overload the target's system with requests. In
this study, we introduce a novel approach to detecting the DDoS attacks in an Edge-IIoT
digital twin based generated dataset. The proposed approach is designed to retain the
already learnt knowledge and easily adapt to the new model in a continuous manner
without retraining the deep learning model. Here are some minor comments for improving the quality of the article.

How are digital twins integrated with the Edge-IIoT framework in this study, and what specific characteristics of digital twins are leveraged for DDoS detection?

Can you provide details about the Edge-IIoT digital twin-based dataset used for DDoS detection? What types of features are included, and how are they relevant to distinguishing between normal and attack traffic?

What specific machine learning or deep learning techniques are used to detect DDoS attacks in the Edge-IIoT environment? How does the proposed approach differ from traditional DDoS detection methods?

The models M1, M2, and M3 achieved accuracy scores of 0.955, 0.960, and 0.966, respectively. How do these models compare with other existing models for DDoS detection, and were any baseline models used for benchmarking?

How well does the approach handle variations in attack patterns or new types of DDoS attacks? Is there a mechanism to dynamically adapt to emerging threats?

Aside from accuracy, what other evaluation metrics were considered (e.g., precision, recall, F1 score) to assess the model’s effectiveness? Are there any trade-offs in terms of detection speed or computational cost?

How scalable is the proposed model in an Edge-IIoT environment? Can the model effectively perform real-time DDoS detection without compromising system performance?

Given the Edge-IIoT environment, how does the study address data security and privacy concerns, especially in handling sensitive information from digital twins?

What are the limitations of the current approach, and what further enhancements are suggested for improving DDoS detection accuracy and adaptability in Edge-IIoT settings?

·

Basic reporting

The paper is written in clear and professional English, making it easy to follow. The author provides a novel approach using deep learning to detect DDoS attacks in an Edge-IIoT digital twin-generated dataset. However, while the dataset is introduced with details like sample counts, class labels, and balancing techniques, the attributes or features are only briefly mentioned. It is unclear whether each sample represents a single time-point data or a time-series of network attributes. The connection between the dataset and the CNN baseline model is not well established, causing confusion about its application. The paper includes appropriate literature references and is generally well-structured, but it would benefit from a more detailed discussion of the dataset's attributes and the CNN's role.

Experimental design

The research is original and falls within the journal's scope, focusing on continuous adaptation of deep learning models for DDoS detection without retraining. The research question is relevant, and the author explains the process through a series of deep learning models trained on different dataset parts. However, more clarity is needed on how DDoS knowledge shifts over time and how this is reflected in the dataset. The use of oversampling for the less-represented class (Normal) raises concerns about potential data leakage, as the test set might contain oversampled Normal samples from the training set. These methodological details need to be addressed to ensure replicability and reliability.

Validity of the findings

The paper successfully proposes a series of models transferring knowledge sequentially, but some concerns regarding the dataset and knowledge shifting need clarification. The author should provide more insight into how the dataset reflects DDoS knowledge shifts over time and address the risk of data leakage due to oversampling. The conclusions are generally well stated and linked to the research question, but they rely on resolving these dataset-related issues for full validity. Additionally, Table 6 contains a typo ("ConvID 4" should be "Conv1D"), which should be corrected for accuracy.

Additional comments

Overall, the author presents a promising approach to DDoS detection using deep learning models with continuous adaptation. To enhance the paper's impact, it is important to address the concerns about dataset attributes, the connection to the CNN model, and the handling of data leakage risks.

---

## Round 0.2 · Minor Revisions

All concerns raised by the reviewers have been partially addressed; the paper still needs further work regarding the experimental setup and justification for the use of the CNN. These issues require a minor revision. If you are prepared to undertake the work required, I would be pleased to reconsider my decision. Please submit a list of changes or a rebuttal against each point that is being raised when you submit your revised manuscript.

·

Basic reporting

The paper is written in clear and professional English; however, there are significant areas that require improvement. While the inclusion of the dataset features in the appendix table is a step forward, inconsistencies in the data types of certain features, such as "tcp.ack" and "dns.qry.type," remain problematic. These features are represented as "float64," which contradicts their names and raises questions about their numerical representation and usage in classifiers. Additionally, features like "icmp.unused," "http.tls_port," and "dns.qry.type" have a mean and standard deviation of zero, indicating they do not contribute to classification and should be removed. Although the author mentions a reduction in features from 64 or 63 to 61, further clarification on this point is necessary. The uneven scale of features, with some having standard deviations of 1e9 while others are around 1e-1, could hinder the CNN's learning process, suggesting that normalization is essential.

Experimental design

The research remains original and relevant to the journal's scope, but the justification for using CNNs in this context is still unconvincing. The author claims that CNNs can capture spatial correlations in network traffic features, yet these features are discretely calculated and lack a fixed spatial order, which is a fundamental characteristic of CNN applications. While the author has provided additional explanations in the rebuttal letter, the argument for CNN suitability is not adequately supported.

Validity of the findings

The findings of the paper introduce a novel approach to DDoS detection, but there are opportunities for further enhancement regarding the assessment of impact and novelty. Ensuring that the underlying data is robust and statistically sound is essential, and addressing the issues related to feature representation and the potential removal of certain features will strengthen the validity of the conclusions. While the conclusions are generally well articulated, they would benefit from a clearer connection to the dataset-related concerns and the appropriateness of the CNN model. Additionally, correcting the persistent typo "ConvID" in Table 6 (first row) to "Conv1D" would improve the overall accuracy of the paper.

Additional comments

In summary, the author has made some commendable improvements since the first submission, but there are still important areas that could be further refined. Clarifying the dataset features, and providing a stronger justification for the use of CNNs will enhance the paper's credibility and contribution to the field of DDoS detection. By focusing on these aspects, the author can significantly strengthen the overall impact of the research.

---

## Round 0.3 · accepted · Accept

I am pleased to inform you that your work has now been accepted for publication in PeerJ Computer Science.

Please be advised that you cannot add or remove authors or references post-acceptance, regardless of the reviewers' request(s).

Thank you for submitting your work to this journal. I look forward to your continued contributions on behalf of the PeerJ Computer Science editors.

With kind regards,

·

Basic reporting

The author has comprehensively addressed all earlier comments. The manuscript maintains clear, professional language and a logical structure. Sections flow smoothly, and key contributions are highlighted effectively.

Experimental design

All major methodological questions have been clarified, with sufficient information on data handling and model procedures. The design is sound and aligns well with the research objectives.

Validity of the findings

Previous errors and typographical issues have been corrected. The findings are now fully supported by the clarified methods and additional validation details. Conclusions are appropriately tied to the presented results.

Additional comments

The author has fully addressed all my concerns, and I have none remaining.